# Development of a Patient Decision Aid for Rectal Cancer Patients with Clinical Complete Response after Neo-Adjuvant Treatment

**DOI:** 10.3390/cancers15030806

**Published:** 2023-01-28

**Authors:** Lien Smets, Annelies Debucquoy, Eva Oldenburger, Chantal Van Audenhove, Lynn Debrun, Jeroen Dekervel, Gabriele Bislenghi, André D’Hoore, Albert Wolthuis, Karin Haustermans

**Affiliations:** 1Department of Radiation Oncology, Universitair Ziekenhuis Leuven, 3000 Leuven, Belgium; 2LUCAS KU Leuven Center for Care Research & Consultancy, 3000 Leuven, Belgium; 3Department of Abdominal Surgery, Universitair Ziekenhuis Leuven, 3000 Leuven, Belgium; 4Department of Digestive Oncology, Universitair Ziekenhuis Leuven, 3000 Leuven, Belgium

**Keywords:** rectal cancer, decision aid, shared decision making, patient education

## Abstract

**Simple Summary:**

Rectal cancer patients with a clinical complete response (cCR) after neo-adjuvant treatment may be spared the surgical morbidity and mortality of radical surgery without oncological compromise. Therefore, discussing treatment options, including organ-sparing strategies (watch-and-wait) as an oncological equivalent alternative to major surgery, is important in the shared decision making (SDM). SDM may be facilitated by a patient decision aid (PtDA). The aim of this study was to develop and evaluate a PtDA for rectal cancer patients who have a cCR after neo-adjuvant treatment.

**Abstract:**

Surgery is the primary component of curative treatment for patients with rectal cancer. However, patients with a clinical complete response (cCR) after neo-adjuvant treatment may avoid the morbidity and mortality of radical surgery. An organ-sparing strategy could be an oncological equivalent alternative. Therefore, shared decision making between the patient and the healthcare professional (HCP) should take place. This can be facilitated by a patient decision aid (PtDA). In this study, we developed a PtDA based on a literature review and the key elements of the Ottawa Decision Support Framework. Additionally, a qualitative study was performed to review and evaluate the PtDA by both HCPs and former rectal cancer patients by a Delphi procedure and semi-structured interviews, respectively. A strong consensus was reached after the first round (I-CVI 0.85-1). Eleven patients were interviewed and most of them indicated that using a PtDA in clinical practice would be of added value in the decision making. Patients indicated that their decisional needs are centered on the impact of side effects on their quality of life and the outcome of the different options. The PtDA was modified taking into account the remarks of patients and HCPs and a second Delphi round was held. The second round again showed a strong consensus (I-CVI 0.87-1).

## 1. Introduction

Surgery is the primary component of curative treatment for patients with rectal cancer, either on its own for early stage disease or after neo-adjuvant chemoradiotherapy for locally advanced disease [1,2]. The current treatment approach is associated with good oncological outcomes [3,4,5,6]. However, radical surgery is also associated with substantial morbidity such as poor anorectal, urinary, and sexual function, affecting quality of life, let alone other short-term surgical complications [7,8,9,10]. Several studies have shown that pathological complete response (ypCR) after neo-adjuvant treatment is affiliated with favorable oncological outcome compared to those without [11,12,13,14,15]. Therefore, one might wonder if radical surgery is absolutely necessary in patients who reach a clinical complete response (cCR) after neo-adjuvant treatment [16,17,18,19,20,21]. With an organ-sparing approach, these well-responding patients may be spared the surgical morbidity and mortality of radical surgery without oncological compromise. Therefore, discussing the different treatment options, including a watch-and-wait (active surveillance) approach as an equivalent alternative to major surgery, is important in shared decision making (SDM).

SDM is defined as a process through which clinicians and patients make healthcare choices together, representing the essence of patient-centered care [22,23]. SDM aims to enable individuals to act independently and to make their own free choices by providing clinical evidence and supporting the patient to deliberate about their options [24]. Additionally, there is growing evidence that patients desire greater autonomy and prefer to be more involved in treatment decision making [25]. The active participation of patients in making decisions not only leads to better clinical outcomes, but also promotes satisfaction and treatment adherence [26]. To participate in SDM, patients need to be well informed about all available treatment options and be able to express their personal values and preferences in consultations with clinicians. Decision support tools, such as a patient decision aid (PtDA), can facilitate SDM [27].

A PtDA is a valuable tool providing information on different treatment options to help patients make informed choices, taking into account their personal values and preferences by asking them to consider their personal pros and cons regarding each treatment option [28,29]. PtDAs explicitly state the decision to be made and provide patient-friendly information on options, outcomes, risks, and benefits in a way that allows the patient to clarify what matters most to them. This way, a PtDA contributes to shared decision making [30]. Previous research has shown that the use of PtDAs improves patients’ knowledge of the available options, allows patients to feel more informed and know what matters most to them, increases patient participation in decision making, and results in more realistic expectations of treatment outcomes [31]. In summary, PtDAs help clinicians and patients make high-quality decisions, that are guided by patient values and take into account the potential tradeoffs in benefits and risks of different options. Unfortunately, their implementation in routine clinical practice is hampered by various barriers, such as lack of training in SDM and concerns that counseling will take too much time [32]. Therefore, a user-centered design to develop the PtDA is proposed to make them more suitable for implementation. This design process involves end-users (healthcare professionals and patients) in the development process, allowing them to provide feedback both on content and design of the prototype, resulting in an improved acceptance among healthcare professionals (HCPs) [33].

To our knowledge, there are no PtDAs available to assist patients making a treatment decision regarding surgery versus active surveillance in rectal cancer. The goal of this study was to develop a PtDA based on user-centered design principles. The developed PtDA aims to provide accurate and balanced information to help patients make an informed, value-based decision when considering treatment options in rectal cancer.

## 2. Materials and Methods

We used a systematic development process that focuses on user-centered design to develop and evaluate the decision aid in two phases: (1) the development of a draft prototype for the decision aid based on a literature review and the knowledge of experts in the field; (2) alpha testing (testing by people directly involved in the development process) of the decision aid to test its comprehensibility, acceptability, and usability (Figure 1) [34].

Participants were former rectal cancer patients; HCPs were specialized in rectal cancer (physicians, nurse specialists and psychologists). Patients were eligible if they were former rectal cancer patients with primary rectal cancer adenocarcinoma who had reached a clinical complete response after neo-adjuvant treatment and age ≥18 years. The patients who matched the inclusion criteria were approached during follow-up to introduce them to the study. If the participant agreed, they were contacted by the researcher to provide more information about the study and its procedure, and to schedule the interview. For HCPs, the inclusion criteria were being employed as a resident, staff member, nurse specialist, or psychologist at the department of radiation oncology, abdominal surgery, or digestive oncology, and specialized in rectal cancer. The HCPs were recruited via email, including the study description and link to the online survey. The study was approved by the research ethics committee of UZ/KU Leuven (ethics code: S65370), and informed consent was obtained from all participants.

First, we developed a draft prototype PtDA based on a literature review and the Ottawa Decision Support Framework, which is specifically relevant for preference-sensitive decisions and meets the quality criteria set out in the International Patient Decision Aid Standards (IPDAS). Furthermore, ‘The Ottawa Decision Support Tutorial: Improving Practitioners’ Decision Support Skills was followed by the investigator and the certificate was obtained. We worked through the key elements of the ODSF to develop the content of the PtDA, clarify the decision by providing facts and probabilities, and describe potential benefits and risks and clarify values. This involves formulating information about the problem, alternatives, and related benefits and risks to improve knowledge on the decision. By defining probabilities tailored to the patient’s clinical risk, more realistic expectations or subjective judgements of the likelihood of benefits and risks are created. Lastly, personal values are clarified by asking individuals to consider the personal importance they place on each benefit and risk and identify trade-offs they will need to make in choosing one alternative. The PtDA prototype was reviewed by one surgeon, one oncologist, and one radiation-oncologist to evaluate the scientific content.

Second, a qualitative study was performed with both an expert panel of healthcare professionals (HCPs) and with patients to review and evaluate the PtDA. A two-step Delphi procedure was used to obtain a consensus on the content of the decision aid. They were asked to evaluate the content of the decision aid on accuracy, comprehensibility, clarity, and relevance. The items were evaluated using a 4-point Likert scale. The participants were asked to rate each item (totally disagree, disagree, agree, strongly agree) and were also given the opportunity to provide qualitative feedback on each item evaluated, such as suggestions or additional comments. Three additional questions were added to the survey: (1) Is there any content missing in the decision aid (2) What is the added value of this decision aid (3) Are there reasons not to use the decision aid. The content validity index (CVI) was used to quantify the degree of content validity. The item–CVI was calculated to assess whether the experts find the information in the decision aid relevant, understandable, complete, and correct. The weighting between the number of experts agreeing was made with regard to the total number of participating experts. A criterion proportion was required to establish acceptable content validity. For each criterion, a consensus was considered reached if at least 80% of survey participants ranked the criterion as totally agree/agree, this being a I-CVI of 0.78.

In addition, semi-structured interviews were performed with former rectal cancer patients to explore (1) the perspective of patients on the decision aid, (2) patients’ information needs, and (3) the acceptability by evaluation of the perspective of patients on the flow of the information, its length, amount of information, balance in presentation of information about options, and suitability for decision making. Each participant was given a copy of the PtDA in preparation of the interview. The interviews took place at the hospital or remotely by Skype/Teams, depending on the patients’ preference and their ability to come to the hospital. The interview was guided by a self-constructed interview guide, and contained open-ended questions to invite the participant to answer freely. During the interview, field notes were taken to note overall impressions or general thoughts about responses. All interviews were audio-recorded and transcribed verbatim afterwards. The transcription, analysis, and coding of the interviews was primarily performed by L.S., and were analyzed by the means of open (line-by-line coding), axial, and selective coding [35]. A combination of a deductive and inductive analytical method was used. Based on the interview guide, a coding list was prepared in advance by L.S., in coordination with E.O. In the first phase, one or more keywords were assigned to each fragment in the transcript based on its content, resulting in an initial list of concepts. In the next phase, concepts were merged and categories were developed using the constant comparative method. In the final step, concepts and categories were reviewed for similarities and collapsed into major themes with subthemes. Evolving themes were discussed by L.S. and E.O. regularly, to ensure the themes and concepts were correct and made sense, and that all possible themes and concepts were recognized. There was a strong consensus on coding and themes; disagreements were discussed, and the theme that covered the content best was chosen. Relevant quotes were selected to represent the patients’ points of view. Based on the final analysis, a list was generated of the most common themes and factors relevant to the treatment decisions made by patients.

The data from the interviews and Delphi survey were used to refine a new version of the PtDA.

## 3. Results

### 3.1. Feedback HCPs (1st Round)

The HCPs evaluated the prototype of the PtDA, which was based on a literature review and clinical experience, in the first Delphi round. Fourteen of the twenty-seven participants completed the survey (response rate: 52%). The content validity was excellent (I-CVI of 0.85-1). Consensus was reached for each item evaluated. From the qualitative feedback given by the HCPs, we concluded that the provision of information would ideally consist of simple, jargon-free language, consistent use of terms and appeals, short sentences, and plain language to increase readability and comprehensibility. They also recommended that the illustrations be enlarged. They also felt that not all aspects of treatment such as side effects and the follow-up trajectory were adequately covered in the PtDA. Some noted that certain clinical aspects should be explained more broadly. Finally, they also disagreed with the way some things were phrased.

The HCPs found all necessary information to be available for the patient to be able to make an informed decision. According to the HCPs, the added value of the PtDA includes that (1) it allows the patient to reflect on the possibilities, (2) it encourages patients to think about the future, (3) the patient will be more actively involved in the decision leading to improved therapy compliance and satisfaction, (4) it gives a clear overview of the possibilities and consequences, and (5) it allows patients to reread information at home. There were two reasons not to use the PtDA in clinical practice: the applicability of shared decision making to the practice population and in cases of language barrier. A few participants had concerns that the use of the PtDA would increase the consultation time.

### 3.2. Feedback Patients

Eleven patients—of whom five were female and six were male—were interviewed. Interviews took about one hour to complete. The age of participants ranged from 37 to 73 years. Three of them had finished high school, four had received higher education and four had a university degree. They were all patients at the University Hospitals of Leuven. Based on the predetermined coding list, the interviews were analyzed. The analysis resulted in the emergence of new themes, as well as the merging of existing ones, resulting in four major themes with several subthemes, these being usability, factors in decision making, usage of the decision aid, and active participation (Table 1).

#### 3.2.1. Usability

Overall, the PtDA was thought to be organized logically. The tool was easy to understand and no clarification was needed to be sought on the internet. However, they indicated that some of the medical terms could be confusing for newly diagnosed rectal cancer patients. One patient expressed concerns about the language, believing that not everyone would understand it, due to the long sentences and medical terminology. The use of illustrations made it easier to understand the text. Patients did not take issue with the content of the table, but did have difficulty reading the text in it because of the small font size and small line spacing. They also prefer a paper PtDA compared to an online version because of various reasons: they find it easier to consult, easier to explain to other people what awaits them, and easier to use in discussion with a general practitioner (GP). The interviewer specifically asked: “What information do you think is missing in the PtDA?”. Based on this question, the patients identified several areas that needed expansion or to which further information needed to be added:timepoints of check-ups after surgery and active surveillance;clarification on low anterior resection syndrome (LARS) and when it occurs;information on what happens if the tumor recurs during active surveillance;clear statements that the tool supplements a physicians’ recommendation;the inclusion of a person’s contact information for additional questions and concerns.

In general, patients indicated that the information provided is understandable, clear, objectively presented, and gives a clear overview of what the options entail and their possible consequences.

#### 3.2.2. Factors in Decision Making

All patients indicated they wanted to be as fully informed as possible. The PtDA makes it very clear that if the tumor is no longer detectable, different treatment options are available, and it gives a very clear representation of the decision to be made. The clear overview of options and consequences would positively contribute to the decision making. The fact that the PtDA involves targeted information will make it easier to make a decision. They reported having missed this tool when faced with choices regarding their treatment. The patients also emphasized the importance of guidance by an HCP in the decision making. The decision aid was found to be useful for supplementing their doctors’ explanations. Patients indicate that having confidence in the physician plays a great role in decision making. Lastly, the information about the acute and late side effects of both treatment options allows patients to assess their potential impact on their quality of life.

#### 3.2.3. Usage of the Decision Aid

The patients reported that a lot of information is given during outpatient clinics, much of which is lost. The main reasons for this were: there is too much information to process and being emotional or feeling overwhelmed by all the information. The PtDA gives the opportunity for patients to reread the information at home. They would also use the decision aid to explain their medical situation to family and friends. All the patients agreed that the PtDA should be used during the consultation with the HCP explaining the treatment options to provide more structure, followed by another consultation a few days later to discuss concerns and the final decision, giving the patient a chance to review the decision aid. The patients noted that additional explanations in the PtDA may be useful if something is not clear to the patient. In addition, they would feel better informed of the potential drawbacks of the different treatment options, as well as able to ask more focused questions in consultations after reading the PtDA. The patients indicated that they prefer to be handed information rather than having to search for it themselves on the internet, which comes with the risk of receiving incorrect information. Patients who are not able to use the decision aid need to be supported by their physician or a nurse specialist in addition to their family, to make sure they have all the necessary information for decision making. An additional consultation with the treating physician may be a good alternative for those patients who are not able to use a PtDA. The patients disagreed on what time best to deliver the PtDA, i.e., at diagnosis or after restaging.

#### 3.2.4. Active Participation

The minority would leave the decision entirely to the physician, because they trust their expertise and do not want to bear the responsibility. Most patients reported that they would like to be involved in the decision. When it comes to making a final decision, the patients want to do so in agreement with the physician. Having as much information as possible was found to be very important in order to participate in the decision making. Open communication also makes patients feel that decisions are not being made on their behalf.

### 3.3. Feedback HCPs (2nd Round)

The PtDA was adapted, taking into account the remarks of both HCPs and patients. A second Delphi round was held to evaluate the prototype 2 before beta testing (field test with patients and HCPs). The HCPs were provided with an overview of comments from the first round, the remarks from patients, and the adjustments made. In the second round, 15 of the 27 participants completed the survey (response rate: 55%). Again, a strong consensus was reached for each item evaluated, with an excellent item–content validity index of 0.87-1. No qualitative feedback was given in the second round.

### 3.4. Final Decision Aid

In the end, the PtDA consisted of an introduction on the choice to be made and the purpose of the PtDA, clarification of the treatment options, additional information on LARS, an overview of the advantages and disadvantages of both options, an overview of the oncological outcomes and functional outcomes of both options, and clarification of the patients’ personal values (Table 2). A non-validated English translation of the original PtDA (Dutch version) is provided as a Appendix A.

## 4. Discussion

In summary, the results of this study helped us to create a PtDA to be tested further in the next phase of our research on decision making in rectal cancer patients with a clinical complete response after neo-adjuvant treatment. Overall, patients found the developed PtDA valuable for future rectal cancer patients in this situation, with most patients expressing how they wished such a tool would have been available to them. Additionally, the HCPs felt that this tool contributes to inform patients about the pros and cons of available treatment options and it could add value to the decisionmaking process.

Partly due to the optimization of neo-adjuvant treatment for rectal cancer, patients are more likely to achieve a clinical complete response. Therefore, discussing treatment options, such as a watch-and-wait approach as an alternative to major surgery, is more common in clinical practice. In addition, various studies offer a watch-and-wait approach if a complete response is achieved [36,37], [NCT04246684]. Patients should be well informed and able to think about different treatment options to reach an informed preference towards a treatment. In conclusion, a lot of patients could benefit from the decision aid to support them in the decision making.

For the development and evaluation of the PtDA, we used a two-step testing approach following the procedure by Coulter et al. [34]. In this manuscript, we report on the first phase (alpha testing) of the study. Alpha testing aims to evaluate the PtDA’s acceptability and usability from the patients’ and HCPs’ perspectives. This alpha testing process, during which the patients and HCPs critique and provide feedback on the comprehensibility and acceptability of the PtDA, assures the validity and reliability of the development process and the final product [34]. The next phase (beta testing) will involve field testing of the PtDA using patients facing the actual decision and HCPs who are interacting with them. The feasibility, satisfaction, and effectiveness of the PtDA will also be assessed during this phase.

The semi-structured interviews provided valuable input for the revision of the PtDA, but there are also limitations. First, the interviewees were all former rectal cancer patients who did not undergo surgery. We did not include patients who did not have a cCR after their neo-adjuvant treatment. This was deliberately so, because patients without a cCR after neo-adjuvant treatment who had undergone surgery might perceive the PtDA as being confrontational, since they missed out on that choice. One of the strengths of the study is that we included a wide range in both the age (37 to 73 years) and the education levels of the patients. Therefore, we assume the content of the PtDA will be easy to understand for users with varying backgrounds.

We initially planned focus groups to evaluate the PtDA by HCPs, but this approach was not feasible due to practical reasons. As an alternative, we chose a Delphi survey with the option to provide additional comments. As the Delphi survey was anonymous, it is impossible to determine whether the same people participated in both rounds.

A concern among a few HCPs was that the PtDA would increase the consultation time, which has been known as one of the major barriers in the implementation of SDM and PtDAs [38]. However, our PtDA is intended to be used at home by the patient to review information, rather than during consultation with the clinician, thus not directly interfering with consultation time. This will allow them to discuss their treatment options in a second consultation with the clinician. However, the majority of the patients noted that the PtDA could provide more structure to the consultation, which may lead to a shorter consultation time after a while. There is inconclusive evidence on the effect of PtDAs on consultation length [30]. We will evaluate the impact of our PtDA on the length of consultation time in the beta testing phase.

Since patients often say ‘you may decide’, HCPs are concerned that patients will not take part in the SDM process. Therefore, we also evaluated whether patients would prefer to be actively involved in the decisionmaking process. Most patients indicated they would like to be involved, but strongly rely on the advice of the clinician. They also noted that the use of the PtDA would lead to more involvement, and would allow them to ask more specific questions and express their concerns. This is consistent with the findings of previous studies [39,40].

## 5. Conclusions

In this study, we followed a user-centered design process to develop a PtDA for patients with rectal cancer. Both former rectal cancer patients and healthcare professionals participated in the development process, which resulted in a PtDA that meets the needs of both HCPs and patients and supplies balanced information on the two available treatment options. The effectiveness and implementation of this tool will be studied in future research.

## Figures and Tables

**Figure 1 cancers-15-00806-f001:**
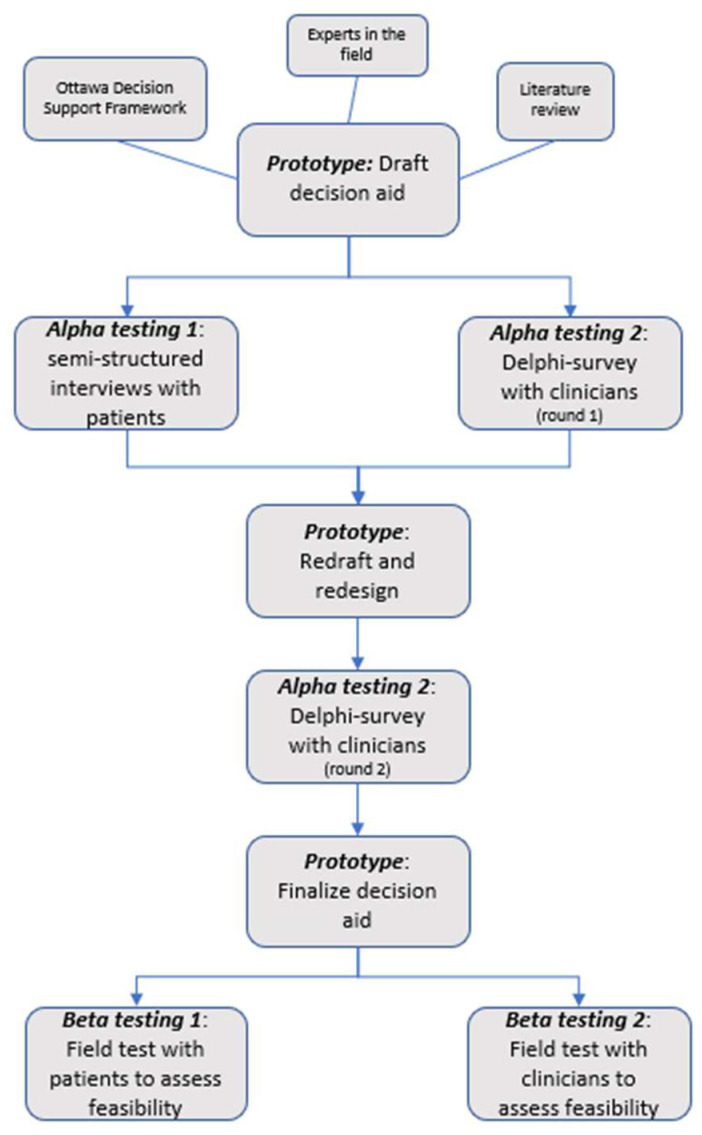
Development process.

**Table 1 cancers-15-00806-t001:** Themes.

Major Themes	Subthemes	Quotes
Usability	Completeness/missing information Comprehensibility Format Neutrality of information	“I think the tool gives a good overview and a good a presentation of the pros and cons.” “Give more information on what follow-up entails, which types of examinations and at what time points’. “It explains very clearly what the consequences of both options are.” “The more images, the better. An image says more than words.” “I think the information is neutral and also keeps the options open.”
Factors in decision making	Information needs Personal arguments Support in decision making (both from healthcare professionals and family/friends)	“I believe that based on what I have read, one will be able to make a decision.” “This is really targeted information. The more information you get, the better you can judge the situation.” “The people you are surrounded by, such as your partner, children, or parents. These are things you discuss together.”
Usage of the decision aid	Timing Motivation Guidance	“I would offer it after neo-adjuvant treatment, when one is effectively faced with the choice.”
		“Such a tool can bring more structure the consultation.” “If you are overwhelmed by emotions or anxiety, you absorb less information. Therefore, it would be helpful that you can reread all the information quietly at home.”
		“The physician can give additional explanation if some information is unclear.”
Active participation	/	“Most of all, I think you will follow the doctors’ advice.” “The decision should not be made over my head. Additionally, for that not to happen, information is very important. If you get enough information, you do not have the impression that decision are made for you.”

**Table 2 cancers-15-00806-t002:** Content decision aid.

Content Decision Aid “Surgery versus Active Surveillance”	
Introduction	What is a decision aid? Aim of the decision aid
What is rectal cancer?	Rectal cancer
Treatment options	Option 1: Surgery Option 2: Active surveillance
Additional information	LARS
Survey	Advantages Disadvantages Oncological outcome Functional outcome
Personal considerations	Questions about personal values to assist patients in the decision making
Conclusion	Possibility to indicate the patient’s preferred treatment option
Find out more?	Interesting websites Peer groups
Questions?	Contact information

## Data Availability

The data are not publicly available.

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
