# Peer review of "Development of a Patient Decision Aid for Rectal Cancer Patients with Clinical Complete Response after Neo-Adjuvant Treatment"

_cancers, 2023, doi:10.3390/cancers15030806_

Round 1

Reviewer 1 Report

As written in the summary „the aim of the study was to develop and evaluate a patients decision aid (PtDA) for rectal cancer patients who have a clinical complete response (cCR) after neo-adjuvant treatment.“  The topic fits the times because due to the prevalence of total neoadjuvant treatment for rectal cancer we expect more patients with cCR suitable for non-operative management in the future.  

The manuscript is well structured and divided into several sections. The key sentence of the „introduction“ is that at the moment „there are no PtDAs available to assist patients making a treatment decision regarding surgery versus active surveillance“. The development process described in „materials and methods“ is illustrated in a figure.

Unfortunately some sections are very narrative and contain few exact data. Other paragraphs could be even more detailed. It is a pity that in the end the PtDA itself is not provided, or at least extracts of it. The reader would have expected that from the topic.

Specific comments:

Lines92/93: how many HCPs were involved? What were their selection criteria? Further details should be provided concerning the participating HCPs.

Line165: 27 patients were asked to participate, 14 completed the survey, the remaining 13 did not complete the survey or did not want to participate at all? Could you please provide details on that.

Line185: Finally 11 patients were interviewed. Could you please clarify the exclusion of another three patients.

Lines260/265: that paragraph concerning round 2 with clinicians is kept very short compared with   the foregoing sections. The sentence „in the second round (response rate 55%) again a large consensus was reached“ is incomprehensible without background informations. Why only 55% response rate for participating healthcare professionals?

Line295: „we did not include patients who did have a cCR after their neo-adjuvant treatment“: in lines 94/95 you included only patients with cCR. Could you please adjust that.

Line296: „because patients without a cCR after neo-adjuvant treatment“: it should be clarified that these patients had undergone surgery.  

Reviewer 2 Report

The present manuscript is dealing with the PtDA for rectal cancer in order to facilitate a SDM. This is a very important step as in patinets with complete clinical response after neoadjuvant treatment, the ideal management can not yet be formulated and there are different alternatives going from follow-up to resection. The methodology is good eventhough the included patient group small. Moreover, the manuscript is well written. The main point however, is missing: how does the PtDA look like and with kind of information has been included? Because this is essential and if not optimal, may mislead the patient.

Were the key issues part of it (see below?)

-        Neoadjuvant treatment is very much depending on the localization of the tumor, in particular relevant for the lower rectal third to avoid severe LARS and to preserve the anal sphincter.

-        Most pivotal diagnostic tool is the MRI which however, has an observer variability for positive lmyph nodes, the main criteria for neoadjuvant treatment.

-        LARS is diagnosed in one third of the patinets treated by radiochemotherapy without surgery

-        The patients should be offered a PtDA before TNT and not only after the compete response as in many situations, e.g. tumors in thee middle third, cT3a,b – surgery alone would be sufficient for cure.

-        One fourth of all patinets will have a re-growth necesitating surgery.

-        Ongoing studies- mentioned, e.g. ACO/ARO/AIO 18.2?

Round 2

Reviewer 1 Report

Dear authors,

thank you for your improvement of the manuscript. As this is an international journal and the title is "Development of a Patient Decision Aid ..." an English version of the PtDA should be provided in addition even if it is not validated.

Author Response

Dear reviewer,

Thank you for your improvement of the manuscript. As this is an international journal and the title is "Development of a Patient Decision Aid ..." an English version of the PtDA should be provided in addition even if it is not validated.
We thank the reviewer for their re-evaluation of our manuscript.
We have added a non-validated version of the PtDA as supplementary material.

Kind regards,

Lien Smets

Reviewer 2 Report

Please add an English version (even if not validated) of the PtDA, not only as supplementary file but also the main components as Table into the text otherwise it is difficult to put your results into the context. 

Author Response

Dear reviewer,

Please add an English version (even if not validated) of the PtDA, not only as supplementary file but also the main components as Table into the text otherwise it is difficult to put your results into the context. 
We thank the reviewer for their re-evaluation of the manuscript.
We have added the non-validated PtDA as supplementary material and a table in the text to give a survey of the the content of the PtDA.

Kind regards,

Lien Smets